# The UV Dose Used for Disinfection of Drinking Water in Sweden Inadequately Inactivates Enteric Virus with Double-Stranded Genomes

**DOI:** 10.3390/ijerph19148669

**Published:** 2022-07-16

**Authors:** Fredy Saguti, Marianela Patzi Churqui, Inger Kjellberg, Hao Wang, Jakob Ottoson, Catherine Paul, Olof Bergstedt, Heléne Norder, Kristina Nyström

**Affiliations:** 1Institute of Biomedicine, Department of Infectious Diseases, University of Gothenburg, 413 46 Gothenburg, Sweden; fredy.saguti@gu.se (F.S.); marianela.patzi.churqui@rheuma.gu.se (M.P.C.); hao.wang@gu.se (H.W.); helene.norder@gu.se (H.N.); 2Göteborgs Stad Kretslopp och Vatten, 424 23 Gothenburg, Sweden; inger.kjellberg@kretsloppochvatten.goteborg.se (I.K.); olof.bergstedt@kretsloppochvatten.goteborg.se (O.B.); 3Department of Clinical Microbiology, Västra Götaland Region, Sahlgrenska University Hospital, 413 46 Gothenburg, Sweden; 4Department of Risk and Benefit Assessment, Swedish Food Agency, 751 26 Uppsala, Sweden; jakob.ottoson@slv.se; 5Applied Microbiology, Department of Chemistry, Lund University, 221 00 Lund, Sweden; catherine.paul@tvrl.lth.se; 6Water Resources Engineering, Department of Building and Environmental Engineering, Lund University, 221 00 Lund, Sweden; 7Department of Architecture and Civil Engineering, Water Environment Technology, Chalmers University of Technology, 412 96 Gothenburg, Sweden

**Keywords:** ultraviolet light, drinking water, enteric viruses, human adenovirus 2, rotavirus SA11, echovirus 30

## Abstract

Irradiation with ultraviolet light (UV) at 254 nm is effective in inactivating a wide range of human pathogens. In Sweden, a UV dose of 400 J/m^2^ is often used for the treatment of drinking water. To investigate its effect on virus inactivation, enteric viruses with different genomic organizations were irradiated with three UV doses (400, 600, and 1000 J/m^2^), after which their viability on cell cultures was examined. Adenovirus type 2 (double-stranded DNA), simian rotavirus 11 (double-stranded RNA), and echovirus 30 (single-stranded RNA) were suspended in tap water and pumped into a laboratory-scale Aquada 1 UV reactor. Echovirus 30 was reduced by 3.6-log_10_ by a UV dose of 400 J/m^2^. Simian rotavirus 11 and adenovirus type 2 were more UV resistant with only 1-log_10_ reduction at 400 J/m^2^ and needed 600 J/m^2^ for 2.9-log_10_ and 3.1-log_10_ reductions, respectively. There was no significant increase in the reduction of viral viability at higher UV doses, which may indicate the presence of UV-resistant viruses. These results show that higher UV doses than those usually used in Swedish drinking water treatment plants should be considered in combination with other barriers to disinfect the water when there is a risk of fecal contamination of the water.

## 1. Introduction

Waterborne viruses are frequently the cause of worldwide outbreaks of viral gastroenteritis [1,2,3]. The introduction of fecal materials due to inadequate treatment of water intended for drinking is the most common source of viral contamination [4,5]. Drinking water treatment plants (DWTPs) in Nordic countries (Norway, Sweden, and Finland) use various disinfection methods, such as chlorination, ozonation, ultrafiltration (UF), and/or UV irradiation [6]. The drinking water treatment process is commonly initiated by separating small particles from raw water through coagulation and flocculation, followed by sedimentation of the flocs and sand filtration to remove particles and microorganisms [6]. UV irradiation is increasingly used at DWTPs due to its efficiency in inactivating a wide range of viruses, such as adenovirus and rotavirus and other pathogens, including oocysts of *Cryptosporidium* and cysts of *Giardia*, which are highly resistant to chlorination [7]. Additionally, unlike chemical disinfectants, such as chlorine and ozone, lower doses of UV irradiation produce limited amounts of disinfectant by-products in the drinking water [8].

UV light inactivates microorganisms by causing photoproducts, mostly pyrimidine dimers in the microorganism genomes [9]. Virus susceptibility to UV irradiation depends, apart from its genomic composition, also on factors such as the structure of the viral particle [10,11,12]. For UV inactivation of viruses in water, some studies have compared dsRNA (rotavirus) with ssRNA mammalian viruses (enteroviruses and caliciviruses) [11,13], while others have used different bacteriophages as surrogates for enteric mammalian viruses [14,15]. Double-stranded DNA (dsDNA) viruses, especially adenoviruses, have been shown to be more resistant to UV irradiation than viruses with other genome compositions, such as double-stranded RNA (dsRNA), single-stranded RNA (ssRNA), and single-stranded DNA (ssDNA) viruses [10,15,16].

The inactivation efficiency of the UV light for viruses in drinking water depends on the UV dose applied. In Nordic countries, the UV dose used in drinking water treatment plants ranges from 250 to 400 J/m^2^ with the goal of reducing the viral load 0.75 to 1.75-log_10_ for adenoviruses and 2.5 to 3.5-log_10_ for other viruses [6].

This study was conducted to investigate the minimum UV dose required for drinking water treatment to inactivate three enteric mammalian viruses commonly detected in surface water [17,18,19,20], groundwater [21,22], and drinking water [23,24].

## 2. Materials and Methods

### 2.1. Host Cells and Virus Stocks

Three cell lines, African green monkey kidney cells (Vero CCL-81), African green monkey fetal kidney cells (MA-104), and human lung carcinoma cells (A549 cells), were used for viability testing of the viruses. Vero and A549 cells were grown in Minimum Essential Media (MEM; Gibco, Bleiswijk, The Netherlands) supplemented with 5% fetal calf serum (FCS), 1% L-glutamine, 100 units/mL of penicillin, and 100 μg/mL of streptomycin (Pen-Strep; Gibco, Life Technologies Corporation, Grand Island, NY, USA). MA-104 was grown similarly but in medium 199 (M2154; Sigma Aldrich, St. Louis, MO, USA) with 2.5% FCS. Cells were maintained at 37 °C in a humidified atmosphere of 5% CO_2_. Stocks of echovirus 30 (EV30) (ATCC VR-1660: GenBank: AF311938.1 7.4 kb genome), simian rotavirus (RV SA11) (ATCC VR-1565, 18.5 kb genome), and human adenovirus type 2 (HAdV2) (35.5 kb genome) were grown in monolayers of Vero cells, MA-104 cells, and A549 cells, respectively. HAdV2 (GenBank: KM458627.1) was obtained from Dr. Wang [25].

In order to obtain a large amount of virus, cells were cultured first on 24-well plates and then scaled up to 175 cm^2^ cell culture flasks. For EV30 or HAdV2 infection, the medium was removed from the monolayers that reached 80–90% confluence, and dilutions of the virus in an MEM medium with 2% FCS were inoculated. The infected cells were incubated at 37 °C in an atmosphere of 5% CO_2_, and examined daily until a complete cytopathogenic effect (CPE) was observed. At 5–9 days postinfection, the supernatants were pooled, diluted at 1/100, and added to the ~80% confluent flasks. After complete CPE was observed on the cells of all flasks, supernatants were harvested, pooled, and subjected to one freeze–thaw cycle to release intracellular molecules, followed by centrifugation to remove cellular debris. Virus titers were determined by TCID_50_ and specific quantitative real-time PCR (qPCR). All aliquots of the virus were stored at −80 °C. RV SA11 was propagated similarly, but in a serum-free M199 medium. Briefly, RV SA11 was activated prior to infection with 5 μg/mL trypsin (Gibco) for 30 min at 37 °C; meanwhile, monolayers were washed at least twice with a serum-free M199 medium and then inoculated with the virus at a ratio of 1:1 virus/medium (M199 0.5 μg/mL trypsin). Infected cells were incubated for 1 h at 37 °C to allow adsorption of the viruses to the cells and were gently shaken at 15 min intervals during the incubation time to prevent cells from drying. Thereafter, RV SA11 inoculum was removed and replaced with an M199 medium with 0.5 μg/mL trypsin. At 5 days postinfection, supernatants were harvested and stored at −80 °C.

### 2.2. Viral Infectivity Titers

Tenfold serial dilutions (1–1/10^−7^) of the viruses were inoculated in 10 replicates per dilution on ~80% confluent monolayer cell cultures in 96-well plates (Nucleon Delta Surface, Thermo Scientific, Roskilde, Denmark). The plates were incubated at 37 °C in a humidified 5% CO_2_ atmosphere and observed daily for 4–9 days until cytopathogenic effects were observed. A standard median tissue culture infectious dose (TCID_50_) assay was used to determine the virus titer of each batch of virus. The TCID_50_ titers were determined when 50% of the cell cultures in wells showed full CPE [26].The number of viral particles/mL could be estimated by assuming that one TCID_50_ will produce 0.69 plaque-forming units (PFUs)/mL [27].

### 2.3. UV Light Inactivation

The inactivation of EV30, RV SA11, and HAdV2 with UV irradiation was determined by the exposure of diluted viruses in water to three UV dose levels using a commercial product where UV irradiation was applied to flowing water. The flow was determined by adjusting the pump as close as possible to 0.1944 L/s for reaching 400 J/m^2^ UV, 0.12963 L/s for 600 UV J/m^2^, and 0.0778 L/s for a UV dose of 1000 J/m^2^. The flow varied between 5% and 16% in the different experiments (Table 1).

Briefly, virus suspension was directly diluted in 3 L of tap water (pH 7.0 and 18.5 °C) to obtain an initial concentration of 10^5^ to 10^6^ TCID_50_/mL that would allow detection of a 4-log reduction. The tap water sample used was collected the same day as the experiment. The viral suspensions were pumped with the Eurom Flow TP800P pump (Genemuiden, The Netherlands) through a cylindrical UV reactor apparatus (Aquada 1, Wedeco GmbH, Herford, Germany) containing a monochromatic, low-pressure mercury lamp emitting radiations at 254 nm. The UV reactor, 470 mm in length and 70 mm diameter, was vertically attached to the wall with water flow in one direction. The low-pressure mercury lamp in Aquada 1 was new and had only been used for less than 24 h at the first experiment. It was therefore assumed that its irradiation was 90–100% of the expected capacity. UV irradiance was monitored with a radiometer sensor that had been factory calibrated to alert whether the UV reactor emits less than 70% of the UV irradiation, which is equivalent to 280 J/m^2^ at maximum flow. No alert occurred during the experiments in this study. The exposure time was calculated based on the flow rate of the suspension flowing through Aquada 1 to obtain the required UV dose for each suspension (Table 1). After each experiment, 4 mL of each UV-light-treated and untreated (control) viral suspension was collected and cultured for each virus suspension and analyzed by TCID_50_. Viral inactivation by three UV dose levels based on the flow and on assumed 90–100% efficiency of the UV lamp (400 (range: 355–447), 600 (range: 455–600), and 1000 (range: 824–1027) J/m^2^) was determined (Table 1). The inactivation efficiency of each UV dose was determined by viability testing of tenfold serial dilutions (10^0^–10^−7^) for treated and untreated viruses in four wells with monolayer cells for each dilution. The cell culture plates were followed and examined for CPE for 5 days for echovirus 30, 7 days for rotavirus, and 9 days for adenovirus. The experiments were performed in duplicate for echovirus 30, in triplicate for rotavirus, and in quadruplicate for adenovirus so as to generate more reliable data for each UV dose tested. The TCID_50_ obtained for untreated, and UV treated, virus suspensions was used to calculate log_10_ reduction for each UV dose level.

### 2.4. Detection of Viral Nucleic Acid by qPCR

Real-time qPCR analysis was performed as a control on all water samples subjected to cell culture to confirm the presence of the intended viruses in the assay and to ensure that no viral contamination had occurred. Total nucleic acids were extracted from 1 mL water suspension of all the three viruses before and after UV treatment using a QIAamp Circulating Nucleic Acid Kit (Qiagen, Hilden, Germany) according to the manufacturer’s instructions. Viral nucleic acids were eluted with 150 µL elution buffer.

The qPCR for the detection of each virus was performed on 7300 Fast Real-Time PCR or QuantStudio™ 5 (QS5) Real-Time PCR instruments (Applied Biosystems, Waltham, MA, USA), and all samples were tested in triplicate. The reaction for RNA viruses was performed in a 25 µL reaction mixture containing 5 µL of the extracted nucleic acids, 1 × reaction mix (Invitrogen), 20 U RNaseOUT^TM^ (Invitrogen), 0.5 µL SuperScript^®^ III/Platinum^®^R Taq Mix (Invitrogen), 0.4 µM of each primer, 0.2 µM of probe, and 4 µL of water. The qPCR was initiated with reverse transcription at 50 °C for 30 min, followed by one cycle of 95 °C for 10 min and 45 cycles of 95 °C for 15 s and 55 °C for 1 min. The RT-qPCR for the detection of HAdV2 was also performed on a 7300 Fast Real-Time PCR system. The reaction mixture for HAdV2 was performed in a 20 µL reaction containing 2 µL of DNA, 1 × TaqMan Universal PCR Master Mix (Thermo Fisher, Waltham, MA, USA), 0.5 µM of each primer, 0.4 µM of probe, and 5.2 µL of water. The qPCR was initiated with a cycle of 50 °C for 2 min and 95 °C for 10 min, followed by 45 cycles of 95 °C for 15 sand 55 °C for 1 min, followed by the extension cycle of 60 °C for 1 min. The primers and probes used are listed in the Appendix A, Table A1. Four tenfold serial dilutions (1/10^5^–1/10^8^) of a 2 µg plasmid containing all targeted regions of virus genomes inserted into the EcoRV site of a pUC157 plasmid (pUC57cl; GenScript HK, Ltd., Hong Kong) were used as a positive control in all qPCR analyses. Sterile water (Sigma Life Science) was used as negative control.

### 2.5. Statistical Analysis

Log_10_ reduction was calculated with the following formula:Log_10_ reduction = log_10_ No − log_10_ Nt(1)

No is the concentration of the infectious virus before exposure to UV light, and Nt is the concentration of the infectious virus after exposure to UV light. Analysis of variance (ANOVA) was used to determine the differences between all three UV doses applied for inactivating three different viruses. *p* < 0.05 was considered statistically significant. Bonferroni post hoc *t*-test (*p* < 0.0167) was further used to compare between two UV doses for inactivation efficiency against the virus. Statistical analyses were performed using GraphPad Prism 9 (GraphPad Software Inc., San Diego, CA, USA).

## 3. Results

### 3.1. Viral Inactivation

All three UV doses, 400, 600, and 1000 J/m^2^, were not significantly different in inactivating EV30, a single-stranded RNA virus (*p* = 0.249). There was a 3.6-log_10_ reduction at the lowest UV dose of 400 J/m^2^, and it reached above 4-log_10_ reduction for both 600 and 1000 J/m^2^ UV doses. Both HAdV2 and RV SA11 were >2-log_10_ more resistant to UV irradiation at 400 J/m^2^ than EV30. There was no significant improvement in inactivation efficiency between the two UV doses, 600 and 1000 J/m^2^, observed for any of the three viruses (Table 2).

For RV SA11, a double-stranded RNA virus, a UV dose of 600 J/m^2^ was required for a virus reduction of 2.9-log_10_, while a 400 J/m^2^ UV dose decreased the virus viability by only 1-log_10_. There was no significant difference in viral inactivation efficiency between 600 and 1000 J/m^2^ UV doses (*p* = 0.046 post hoc *t*-test). RV SA11 achieved a 3.3-logs reduction at a 1000 J/m^2^ UV dose with an average initial viral concentration of 7.7 × 10^5^ TCID_50_/mL (Table 2). HAdV2, a double-stranded DNA virus, showed a similar inactivation trend to that for the dsRNA RV SA11. HAdV2 was also inactivated more efficiently in water by a UV dose of 600 J/m^2^ compared with 400 J/m^2^, with a virus reduction of 3-log_10_ versus less than 1-log_10_. Increasing the UV dose to 1000 J/m^2^ did not reciprocate for the significant increase of HAdV2 inactivation (*p* = 0.029 post hoc *t*-test).

### 3.2. Confirmation That the Correct Virus Was Used in the Experiment

To confirm that no contamination had occurred, samples from UV-treated and untreated water/virus suspensions were subjected to qPCR assays for each virus. The virus species in tenfold serial dilutions (1 × 10^0^–1 × 10^7^) of each water sample that was subjected to cell culture was confirmed by qPCR. All expected viruses were identified by qPCR, as shown in the Appendix A, Table A2. The qPCR assay was about 1000 times less sensitive compared with the viability assay in a cell culture for all the three viruses. This difference in sensitivity may be explained by the qPCR having a sensitivity of 10 genomic copies per 5 µL of extracted nucleic acids [28], and the cell culture assay has a detection limit of 1 virus particle per 200 µL dilution.

## 4. Discussion

A UV dose of 500–600 J/m^2^ was shown to be required for a drinking water treatment to efficiently inactivate enteric nonenveloped viruses with double-stranded genomes, such as the adenovirus (dsDNA) and rotavirus (dsRNA) used in this study. Adding known viruses to the drinking water to assess the effectiveness of the UV dose for viral inactivation is not possible on a full scale at a drinking water treatment plant. In this study, we used a UV reactor, providing a full-scale reactor simulation regarding flow rate, UV transmittance, lamp status, and UV intensity. This simulates as much as possible the conditions encountered by a virus in full-scale UV light installations [29]. Globally, different UV doses are used for the inactivation of microbes in drinking water. Since 2003, the Environmental Protection Agency (EPA) in the USA recommends a UV dose of 1860 J/m^2^ to ensure a 4-log inactivation of all viruses, including adenovirus, in drinking water [29]. Previously, a 400 J/m^2^ UV dose was recommended, which still is used in many European countries, including Sweden, and gives a good barrier effect for most pathogenic bacteria and parasites [30,31,32]. In Norway, some drinking water treatment plants use an average of only 300 J/m^2^ [6]. While less than 400 J/m^2^ can be efficient in inactivating bacteria and parasites [7], the results from this study indicate that even 400 J/m^2^ is too low to inactivate at least viruses with double-stranded genomes.

The UV inactivation of picornaviruses with up to 4-log reduction of echoviruses 1, 11, 12, coxsackievirus B5, poliovirus, and hepatitis A virus has previously been achieved by UV doses between 280 and 500 J/m^2^ [7,13,33]. Based on these consistent findings for the UV inactivation of small, spherical, nonenveloped, single-stranded RNA viruses, the UV sensitivity of echovirus 30 in this study did not differ considerably from the previously reported UV radiation inactivation kinetics for other ssRNA enteroviruses. In our study, a 3-log reduction of adenovirus 2 needed a dose between 600 and 1000 J/m^2^, which was at a similar or lower range of the UV dose compared with what had previously been found [34,35,36,37,38,39]. Interestingly, several research studies showed that a UV dose of 400 J/m^2^ achieved a 1-log reduction of adenovirus 2 [33,34,36], which is consistent with our study.

For rotavirus RV SA11 inactivation, a 3-log reduction was achieved at a UV dose of 520–577 J/m^2^ in our study, which is somewhat higher than 280–440 J/m^2^ described in previous studies [16,40]. Variation of rotavirus inactivation results can be subjective to experimental conditions especially suspending media in solutions, such as fresh water, marine water, and phosphate-buffered saline (PBS). Caballero et al. (2004) [41] showed that rotavirus suspended in fresh water needed a higher UV dose of 800–1400 J/m^2^ for 3-log reduction, which could explain the higher doses needed in our study with the virus suspended in tap water. The studies showing lower UV doses for inactivation suspended the rotavirus by phosphate-buffered saline (PBS) [11,16,40]. Discrepancies in virus inactivation results among research studies may be the result of the subjective nature of the TCID_50_ assay, different cell lines, and the UV reactor setup [29]. Most research studies use a bench-scale testing UV reactor, in which UV light is directed down a collimating beam to a petri dish with viral suspension and assesses viral inactivation based on exposure time [29]. This study used an Aquada 1 UV reactor, which provides a full-scale reactor simulation. It uses flow rate to calculate water detention time to UV irradiation, which may explain differences from other studies, but better resembles the method used in DWTPs.

Early knowledge showed that viruses containing double-stranded DNA or RNA genomes are more resistant to UV irradiation than viruses with single-stranded nucleic acid genomes [14,42]. One explanation for the differences in reduction found in this study between the viruses may, to some extent, also depend on the length of the viral genomes. The shorter is a genome, the more nucleotides will be affected by the UV light. However, other factors may play more important roles for UV resistance. Double-stranded DNA or RNA is may be more resistant because of the role played by an additional strand used as a template for replication inside host cells. In this study, we observed a similar trend whereby echovirus 30 (single-stranded RNA genome) was more sensitive to 254 nm UV irradiation as compared with rotavirus SA11 (double-stranded RNA) and human adenovirus 2 (double-stranded DNA). Araud et al. (2020) [11] suggested that 254 nm UV irradiation compromises both virus–host interactions and genome replication of single-stranded RNA, but not host–receptor interaction of rotavirus, a double-stranded RNA virus.

The inactivation process used for drinking water is designed not to destroy all microorganisms but rather to reduce their numbers to an acceptable level. For the double-stranded RNA rotavirus, a reduction to less than 1.1 × 10^5^ viral particles per L of drinking water for tolerable disease burden [43] was obtained at 600 J/m^2^, followed by a plateau with no further reduction at the higher UV dose, where infectious virus was still detected at 1000 J/m^2^. This was also observed for the other viruses. One reason for the high resistance of adenovirus to UV could be the result of nucleic acid damage being repaired during infection [44]. However, as UV resistance was seen for RNA viruses and has been shown in other studies, a plateau of the reduction of viruses at higher UV doses indicates a possible selection of UV-resistant viruses [11,16,40].

The exact UV dose in each experiment could not be determined in this study. However, the detector of Aquada 1 would alarm if the intensity became lower than 70% of the assumed effect and the sensor showed that the UV lamp, with a shelf life of 1 year, had only been used for 1 day and emitted between 90% and 100% of its full capacity during the experiments. Based on this, we could assume that the lamp emitted the expected UV intensity. In addition, we did not use the collimated beam, which is easy for controlling the UV transmission to the surface of the petri dish to 100% with a radiometer displaying the actual UV dose. This may have had implications on our estimated UV doses. However, in our experiment, the virus in the water was exposed to UV light during the water flow more similar to treatment in a water treatment plant than to irradiation in a petri dish.

For all the three viruses used in this study, there were insignificant changes in inactivation efficiency between the two UV doses, 600 and 1000 J/m^2^, which may imply that UV-resistant strains may be common among several viral families. The apparent reason for the persistence of viral response to higher UV doses is unclear but should be taken into account when selecting UV treatment as the only disinfectant in DWTPs. One factor that can affect the development of resistance may be the wavelength of the UV light, which is most effective for inactivating microbes at 253 to 265 nm [45]. In this study, we used UV light at 254 nm, which is the most common wavelength used. However, studies using 220 or 222 nm UV light for the inactivation of rotaviruses and adenoviruses showed a higher reduction at lower UV doses, and no plateau was observed [11,46]. Whether new pulsed UV lamps or different wavelengths of the UV light should be used at the DWTP needs to be investigated to ensure that viruses, which pass through the filtration process, become inactivated before the drinking water is distributed to the consumers.

## 5. Conclusions

Our results indicate that a 400 J/m^2^ UV dose, while known to be efficient for inactivating bacteria, parasites, and bacteriophages with single-stranded genomes, does not appear to inactivate all human pathogenic viruses; 600 J/m^2^ UV was required for considerable inactivation of the viruses with double-stranded genomes. This study used full-scale UV reactor simulation to the actual setup in DWTPs by using Aquada 1 for the inactivation of echovirus 30, rotavirus, and adenovirus 2 suspended in drinking water. The findings from this study highlight the importance of using substantially higher UV doses than those presently used in most DWTPs for significant inactivation of double-stranded DNA or RNA viruses.

## Figures and Tables

**Table 1 ijerph-19-08669-t001:** UV dose for the EV30, RV SA11, and HAdV2 inactivation at various flow rates in tap water with UV transmittance values at 94% (max) and 85% (min).

Virus	400 J/m^2^				600 J/m^2^				1000 J/m^2^			
	Flow Rate (L/s)	Detention Time (s)	UV Dose (J/m^2^)		Flow Rate (L/s)	Detention Time (s)	UV Dose (J/m^2^)		Flow Rate (L/s)	Detention Time (s)	UV Dose (J/m^2^)	
			Max	Min			Max	Min			Max	Min
EV30	0.17	11.28	448	403	0.14	12.75	506	456	0.08	23.50	1011	910
	0.18	10.79	429	386	0.12	15.11	600	540	0.08	23.39	1006	906
RV SA11	0.20	9.94	395	356	0.13	14.19	564	507	0.08	22.60	972	875
	0.20	9.94	395	356	0.12	14.57	579	521	0.08	23.81	1024	922
	0.20	9.94	395	356	0.12	14.57	579	521	0.08	23.81	1024	922
HAdV2	0.19	10.10	401	361	0.12	14.84	590	531	0.08	23.41	1007	906
	0.20	9.94	395	356	0.12	14.84	590	531	0.08	23.34	1004	904
	0.19	10.38	412	371	0.12	14.64	582	524	0.08	23.50	1011	910
	0.19	10.38	412	371	0.13	14.08	560	504	0.08	23.50	1011	910

**Table 2 ijerph-19-08669-t002:** Log_10_ viral reduction of water suspension with EV30, RV-SA11, and HAdV2 after treatment with UV irradiation.

Virus	UV Dose Range (J/m^2^)		Initial Viral Concentration	Log_10_ Reduction			ANOVA
	Min	Max	Mean (TCID_50_/mL)	Mean	Min	Max	(*p*-Value)
EV30	386	448	9.05 × 10^4^	3.60	2.84	4.36	
	506	600	8.70 × 10^4^	4.70	4.20	5.20	
	906	1011	1.39 × 10^5^	5.18	5.00	5.36	0.249
RV SA11	356	395	9.29 × 10^5^	0.95	0.50	1.34	
	507	579	5.99 × 10^5^	2.89	2.00	3.67	
	875	1024	7.70 × 10^5^	3.33	3.00	4.00	0.008
HAdV2	356	412	1.66 × 10^3^	1.04	0.16	2.00	
	504	590	1.36 × 10^4^	3.06	2.00	4.70	
	904	1011	9.17 × 10^2^	2.99	2.70	3.20	0.013

## Data Availability

All data is provided.

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
