# Peer review of "The UV Dose Used for Disinfection of Drinking Water in Sweden Inadequately Inactivates Enteric Virus with Double-Stranded Genomes"

_ijerph, 2022, doi:10.3390/ijerph19148669_

Round 1

Reviewer 1 Report

In this paper, the authors examined the disinfection of UV irradiation on three viruses in spiked water. The authors concluded that the UV dose commonly used in Sweden is insufficient to remove virus with double-stranded genomes. The topic fit well to the aim and scope of the journal. However, the design of the experiment is too simple, particularly that only one model virus for each genome type was examined. Genome type may be not the sole factor affecting the stability of virus. For instance, different shape of virus and capside (protein shell) may also affect the effectiveness of UV irradiation. Therefore, at least three model viruses for each genome type should be considered to verify the conclusion. In addition, the authors only tested virus spiked in tape water, while there may be other factors such as organic matters affecting the effectiveness of UV irradiation. It is questionable whether the current conclusion can be expanded in real scenarios. Furthermore, the authors should also use technology such as electrophoresis to visualize the integrity of virus nuclear acid. Some comments are shown in below,

1.  The result of RT-qPCR should be presented along with those using culture-based method. The author suggested that RT-qPCR is less sensitive compared to culture-based method is somehow counterintuitive. The authors should check the method and use electrophoresis to confirm the nuclear acid extraction efficient. 

2. The result of significance test should be provided in figures or tables. Asterisk or significance letters should be added in the corresponding comparisons. Besides, the figure 1 and table 1 are actually showing exact the result, which is unacceptable in a scientific paper. 

3.  The length of virus nuclear acid should provided. The authors concludes that double stranded genomes is more resistant to UV irradiation. However, it is possible that such difference is result from different length of virus genomes rater than types.

4. Some of the appendix tables (table A1, A2) can be moved to main text. 

5. The graphical abstract is too simple and fails to present the primary findings of the current study.

Author Response

  1. The result of RT-qPCR should be presented along with those using culture-based method. The author suggested that RT-qPCR is less sensitive compared to culture-based method is somehow counterintuitive. The authors should check the method and use electrophoresis to confirm the nuclear acid extraction efficient. 

Response: In the qPCR we detect about 10 RNA or DNA molecules in 5uL extracted material, corresponding to the same volume of virus dilution. In the cell culture we can detect one virus particle in 200uL of virus dilution, which makes the cell culture about 400 times more sensitive than the qPCR even if the qPCR was optimal, which not all of our qPCRs were despite hard effort to get them as sensitive as possible.

  1. The result of significance test should be provided in figures or tables. Asterisk or significance letters should be added in the corresponding comparisons. Besides, the figure 1 and table 1 are actually showing exact the result, which is unacceptable in a scientific paper. 

Response: We thank the reviewer for this comment and we agree that table 1 and figure 1 nearly show the same thing and we have removed figure 1.  We have added the p-values in table 2.

  1. The length of virus nuclear acid should provided. The authors concludes that double stranded genomes is more resistant to UV irradiation. However, it is possible that such difference is result from different length of virus genomes rater than types.

Response: We thank the reviewer for these comments. We have added the length of the viral genomes in materials and methods. Since enterovirus have a shorter genome compared to adenovirus (4.7 times larger) and rotavirus (2.5 times larger), of course more comparable nucleotides of the enterovirus may become affected, but we do not think this can be the whole explanation since even in longer genomes several nucleotides will become affected and the formation of pyrimidine dimers and genome breakage will be lethal to the virus. Other mechanisms as discussed in the manuscript may be as important as the length of the genome. We have added a sentence on this in the discussion part.

  1. Some of the appendix tables (table A1, A2) can be moved to main text. 

Response: We agree and have moved these table to the main text, since they are informative and may be missed as appendix tables.

  1. The graphical abstract is too simple and fails to present the primary findings of the current study.

Response: We thank the reviewer for this comment and have changed the graphical abstract to become more informative showing the findings better.

Reviewer 2 Report

Major concerns:

1.    The authors designed 3 UV doses of 400 J/m2, 600 J/m2 and 1000 J/m2 by control the water flow, this might affect the UV inactivating efficacy as at higher flow rate the viruses accepted shorter time of UV irradiation, indicating by HAdV2 had a less log10 reductions at 1000 J/m2 than those at 600 J/m2 (Figure 1). I thought using UV lamps with different power worked at a constant flow should be better for this study.

2.    EPA in USA recommends a UV dose of 1,860 J/m2 to ensure a 4-log inactivation of all viruses in drinking water. In this study the authors checked the inactivation of 3 viruses at a maximum UV dose of 1000 J/m2. They should check the background of virus contaminations in the drinking water in Nordic countries, e.g., by qPCR, and judge if the 2.9-log to 3.6-log reductions at UV doses of 400 J/m2 presently used in most DWTPs were safe enough for publics.

Major concerns:

1. Title: “The UV dose used for disinfection of drinking water in Sweden does not efficiently …….” might be more appropriate for the manuscript.

2. Line 175: Student’s t-test was not suitable here. ANOVA was used to determine the log10 reduction differences between all three UV doses, adjusted t-test, e.g., LSD-t, Ducan’s t, should be applied to determine the significance of differences between 2 doses, rather than Student’s t-test.

3. Information in Table 1 was duplicated with Figure 1.

Author Response

major concerns:

  1. The authors designed 3 UV doses of 400 J/m2, 600 J/m2 and 1000 J/m2 by control the water flow, this might affect the UV inactivating efficacy as at higher flow rate the viruses accepted shorter time of UV irradiation, indicating by HAdV2 had a less log10 reductions at 1000 J/m2 than those at 600 J/m2 (Figure 1). I thought using UV lamps with different power worked at a constant flow should be better for this study.

Response: We agree that such experiment could have given better and more exact results. Unfortunately we did not have access to such device with a UV lamp with different powers. To better show the results we obtained we did exact measures on the flow and the efficiency of the UV lamp, which is now added in table 1.   

EPA in USA recommends a UV dose of 1,860 J/m2 to ensure a 4-log inactivation of all viruses in drinking water. In this study the authors checked the inactivation of 3 viruses at a maximum UV dose of 1000 J/m2. They should check the background of virus contaminations in the drinking water in Nordic countries, e.g., by qPCR, and judge if the 2.9-log to 3.6-log reductions at UV doses of 400 J/m2 presently used in most DWTPs were safe enough for publics.

Response: We agree that this is an important study that has to be performed.  Earlier work has shown that UV light alone is not enough, but additional purification steps are needed as is mentioned in the manuscript.  

Major concerns:

  1. Title: “The UV dose used for disinfection of drinking water in Sweden does not efficiently …….” might be more appropriate for the

Response: We agree that the tile should be more appropriate and have changed it to:

“The UV dose used for disinfection of drinking water in Sweden inadequately inactivate enteric virus with double-stranded genomes”

  1. Line 175: Student’s t-test was not suitable here. ANOVA was used to determine the log10 reduction differences between all three UV doses, adjusted t-test, e.g., LSD-t, Ducan’s t, should be applied to determine the significance of differences between 2 doses, rather than Student’s t-test.

Response: We have corrected and explained the statistical methods used in the manuscript.

  1. Information in Table 1 was duplicated with Figure 1.

Response: We thank the reviewers for this comment and have removed figure 1.

Reviewer 3 Report

Dear Authors,

Saguti et al. presented a manuscript, which focused on the investigation of the UV dose to inactivation enteric viruses with double stranded genomes. The paper finds that 600 J/m2 UV was required for considerably inactivation of the viruses with double stranded genomes. This finding has a high importance because of using lower UV doses, not for effective inactivation of double-stranded DNA or RNA viruses in most of DWTPs. These findings can be interesting for International Journal of Environmental Research and Public Health readerships. The relevance of the paper is well emphasized. The paper is generally well constructed, the research design is appropriate. The methods are adequately described, and the results are clear presented. There are a few editorial mistakes in the manuscript:

¾     line 43 - space should be between irradiation and [6];

¾     line 48 - the „Cryptosporidium” and „Giardia” should be written in italics.

My suggest is to accept the manuscript titled „The UV dose used for disinfection of drinking water in Sweden does not inactivate enteric viruses with double stranded genomes” in present form.           

Author Response

The paper is generally well constructed, the research design is appropriate. The methods are adequately described, and the results are clear presented. There are a few editorial mistakes in the manuscript:

¾     line 43 - space should be between irradiation and [6];

¾     line 48 - the „Cryptosporidium” and „Giardia” should be written in italics.

My suggest is to accept the manuscript titled „The UV dose used for disinfection of drinking water in Sweden does not inactivate enteric viruses with double stranded genomes” in present form.

Response: We thank the reviewer for the corrections which are now added in the manuscript. We changed the title according to the suggestion of reviewer 2 to : “The UV dose used for disinfection of drinking water in Sweden inadequately inactivate enteric virus with double-stranded genomes” . We hope this change is acceptable otherwise we may change back to the first title.                

Reviewer 4 Report

The article is exceptionally written.

The only concern I have is the authors need to include the ethics statement.

Author Response

Specific comments

 Abstract & Introduction –

  1. The major problem that the study is aiming to solve is not clear in the abstract. The authors should thus consider rephrasing the abstract to bring out clearly the relevance of the study.

Response: We agree that the aim was not clearly given in the abstract, which is now rephrased to pinpoint on the relevance of the study.

Materials and Methods

  1. What was the rationale for performing the experiments in duplicate for echovirus, in triplicate for rotavirus and quadruplicate for adenovirus for each UV dose tested? -The culture results should be presented.

Response: We agree that the cell culture results should be presented and are now given in appendix table A2. The reason for the different number of replicates was to confirm the results, since the high resistance of adenovirus and rotavirus and especially the lack of further declines in viability were surprising for us. We had also made more experiments with echovirus 30, all with the same results, and were therefore not presented.

  1. - More detailed results of the qPCR are needed like: what was the limit of detection of the molecular assays (Ct) to better estimate the check for contamination /is the data presented mean or median?

The qPCR:s were developed and described by Wang et al., 2020, reference 48. They have a sensitivity of 10 genome copies per 5 ul extracted nucleic acid. Since the main aim was to determine the viability of the viruses after UV treatment and only confirm the absence of contamination by qPCR, we only show the presence or lack of qPCR reaction in each dilution analysed.  

  1. -Ethics approval/statement to conduct the study should be included.

Response: Since we only analyse for added stock viruses in drinking water no ethical approval is needed. The study does not have any relevance to any living organism. 

 Results, Discussion and Conclusion

  1. Table 1 is redundant as it is duplication of Figure 1

Response: We agree and figure 1 is removed from the manuscript

  1. -Table A1 can be improved.

Response: The tables have been improved and moved to the main text.

  1. In the paragraph (lines 273-281), the authors have stated a serious limitation of the study in terms of being unable to determine the exact UV dose. Therefore, do the authors think the findings of the study are still credible and still hold? With the highlighted limitation, the conclusion section can be stated with some more caution and therefore should be revised.

The only change that I can see possible

Response: We thank the reviewer fore this comment. We wanted to state that despite this possible limitation the results are reliable. We have rephrased this in the discussion part.

Reviewer 5 Report

Either in the Introduction or Discussion, it should be noted that structurally these enteric viruses are non-enveloped.

The authors did not fully develop and explain the qPCR experiment.  If the purpose was to confirm that no contamination had occurred, what does a negative result tell?  The legend alone does not adequately explain the result and insufficient explanation given in Results 3.2

Line 99             use either by or with, not both

Line 202          tape should be tap

Line 213 –        What is meant by Spiking? Please explain.

Lines 244-245 I think “by” is left out of the following sentence:  suspended the rotavirus phosphate buffered saline

Line 270          “Hower” should be corrected to however

Line 275          “that” should be than

The research investigates which level of irradiation will inactivate enteric water bourne viruses.  Three UV doses were tested (400, 600, and 1000J/m2).

I do feel that the topic is relevant in the field because it included the UV dose used to eliminate pathogens from water treatment facilities in Nordic countries.

The finding is significant in that the findings lead to the recommendation  that higher UV dose (in conjunction with other methods) be used in Swedish water treatment facilities to inactivate enteric viruses.

The methodology is appropriate to measure the reduction in viral titers.The authors determined the TCID50 , which indicated the number of viral particles post treatment. I feel that  The authors did not fully develop and explain the qPCR experiment.  If the purpose was to confirm that no contamination had occurred, what does a negative result tell?  The legend alone does not adequately explain the result and insufficient explanation given in Results 3.2

The conclusions are consistent with the evidence presented. 

The references are appropriate.

The data in each of the tables and figures are clearly presented; the figure legends and table headings are clear with the exception of Table A4.  The authors did not clearly explain how the results presented show that there was no contamination.

Author Response

Either in the Introduction or Discussion, it should be noted that structurally these enteric viruses are non-enveloped.

Response: We thank the reviewer for this comment and this has been added in the introduction part of the manuscript

The authors did not fully develop and explain the qPCR experiment.  If the purpose was to confirm that no contamination had occurred, what does a negative result tell?  

Response: The qPCR:s had previously been developed (Wang et al 2020) and the sensitivity determined to 10 genome copies per 5 ul extracted nucleic acids. A positive result shows that the virus intended was present in the diluted sample. A negative result at low dilution, concentrated viral suspension, would indicate contamination. If the qPCR is negative at a higher dilution and CPE is still present indicates that the qPCR is less sensitive that the cell culture assay which may detect only one virus particle in 200 ul dilution. In each experiment dilutions of a plasmid containing the viral sequences was used to confirm the sensitivity of the qPCR.      

The legend alone does not adequately explain the result and insufficient explanation given in Results 3.2

Response: We thank the reviewer for this comment and have changed the legend and added a sentence on the sensitivity difference between the two assays.

Line 99             use either by or with, not both

Response: This has been changed.

Line 202          tape should be tap

Response: This has been changed

Line 213 –        What is meant by Spiking? Please explain.

Response: We thank the reviewer for this comment, we have changed this expression to adding known virus which is more appropriate expression to use.

Lines 244-245 I think “by” is left out of the following sentence:  suspended the rotavirus phosphate buffered saline

Response: “by” is added in this sentence

Line 270          “Hower” should be corrected to however

Response: This is corrected

Line 275          “that” should be than

Response: This is corrected

 The research investigates which level of irradiation will inactivate enteric water bourne viruses.  Three UV doses were tested (400, 600, and 1000J/m2).

I do feel that the topic is relevant in the field because it included the UV dose used to eliminate pathogens from water treatment facilities in Nordic countries.

The finding is significant in that the findings lead to the recommendation  that higher UV dose (in conjunction with other methods) be used in Swedish water treatment facilities to inactivate enteric viruses.

The methodology is appropriate to measure the reduction in viral titers.The authors determined the TCID50 , which indicated the number of viral particles post treatment. I feel that  The authors did not fully develop and explain the qPCR experiment.  If the purpose was to confirm that no contamination had occurred, what does a negative result tell?  The legend alone does not adequately explain the result and insufficient explanation given in Results 3.2

The conclusions are consistent with the evidence presented. 

The references are appropriate.

The data in each of the tables and figures are clearly presented; the figure legends and table headings are clear with the exception of Table A4.  The authors did not clearly explain how the results presented show that there was no contamination.

Response: We have added information regarding this in the result part and hope this makes it clearer. We want to thank the reviewer for valuable comments and corrections which have improved our manuscript.

Round 2

Reviewer 1 Report

This is a revised manuscript and I found that my comments were carefully addressed.

Reviewer 2 Report

None.